# Live-cell tracking of biliverdin trafficking reveals metabolic exchange between plastids and peroxisomes

Mone Shibata[1,2,*], Keiji Yoshida[1,2,*], Hitomi Takahashi[1] and Yutaka Kodama[1,2,‡]

## ABSTRACT

Metabolite trafficking between organelles is widespread, yet many trafficking routes remain poorly understood. Biliverdin (BV), a tetrapyrrolic pigment, is involved in redox and light signaling in animal and plant cells. To visualize the inter-organelle trafficking of BV in living cells, we used small ultrared fluorescent protein (smURFP), a BV-induced far-red fluorescent protein typically used for deep-tissue imaging in animals. Upon targeting smURFP to various organelles in plant cells, we detected fluorescence from BV-bound smURFP in plastids and peroxisomes. Enhanced BV biosynthesis in plastids led to greater smURFP fluorescence in peroxisomes and vice versa, indicating bidirectional trafficking of BV between these organelles. We also observed trafficking of BV between the cytosol and both plastids and peroxisomes, suggesting that bidirectional trafficking of BV between plastids and peroxisomes occurs via the cytosol. These findings provide a dynamic model for organelle-level metabolite trafficking.

KEY WORDS: *Allium cepa*, Biliverdin, Chloroplast, Cytosol, Far-red fluorescent protein, Fluorescence, Peroxisome, Plastid, smURFP, Tetrapyrrole

## INTRODUCTION

The intracellular distribution and inter-organelle trafficking of metabolites are critical factors influencing cellular metabolism, signaling pathways, and overall cell function. However, our understanding of where and how metabolites are distributed and trafficked within the cell is extremely limited, owing to a lack of appropriate visualization techniques.

The linear tetrapyrrolic pigment biliverdin IXα (BV) is produced by the breakdown of heme, a ring-shaped tetrapyrrole, in a reaction catalyzed by heme oxygenase (HO) (Tenhunen et al., 1968, 1969). In animals, BV is then converted into the strong antioxidant bilirubin by BV reductase (BVR) (Yamaguchi et al., 1994). In addition to being a key intermediate in the production of bilirubin from heme (Tenhunen et al., 1968, 1969), BV has also received attention for its own antioxidant activity (Jansen et al., 2010). In plants, which were originally thought not to produce bilirubin, BV is best known as the substrate for biosynthesis of phytochromobilin (PΦB), the

chromophore precursor of red/far-red photoreceptor phytochromes, through a reaction catalyzed by PΦB synthase (Kohchi et al., 2001). However, we recently discovered that bilirubin is produced through a nonenzymatic reaction of BV with nicotinamide adenine dinucleotide phosphate (NADPH) in the plastids (chloroplasts) (Ishikawa et al., 2023). Therefore, BV is an important precursor not only for PΦB but also for bilirubin in plants.

We performed fluorescence imaging of bilirubin using the bilirubin-induced fluorescent protein UnaG as a proof of concept for intracellular metabolite imaging (Kumagai et al., 2013; Ishikawa et al., 2023; Ishikawa and Kodama, 2024). Using UnaG, we determined that bilirubin is produced in the plastids and is also present in other subcellular compartments such as the cytosol, endoplasmic reticulum (ER), mitochondria, and peroxisomes (Ishikawa and Kodama, 2024). Importantly, when we increased bilirubin levels in plastids through the ectopic expression of plastid-targeted BVR from rat (*Rattus norvegicus*), we observed lower bilirubin levels in peroxisomes (Ishikawa and Kodama, 2024). Given the reciprocal changes in bilirubin levels between these two organelles, we proposed a model in which the plastids share a common pool of the bilirubin precursor BV with the peroxisomes and transport BV to the peroxisomes in plant cells. However, such inter-organelle trafficking of BV remains to be demonstrated.

In this study, we confirmed the trafficking of BV between plastids and peroxisomes using the BV-induced fluorescent protein smURFP (small ultrared fluorescent protein) (Rodriguez et al., 2016), a far-red fluorescent protein that binds to BV as a fluorophore. smURFP is typically fused to a protein of interest as a fluorescent tag and is often used for noninvasive deep-tissue imaging in animals owing to its long excitation (642 nm) and emission (670 nm) wavelengths (Rodriguez et al., 2016). To visualize the smURFP tag for deep-tissue imaging in animals, exogenous BV and BV variants are used as fluorophores (Rodriguez et al., 2016). Here, we incorporated smURFP into a genetically encoded probe to visualize endogenous BV in living cells. Our smURFP-based BV imaging revealed the inter-organelle trafficking route of BV between plastids and peroxisomes in plant cells.

## RESULTS

### smURFP fluorescence originates from endogenous BV in plastids

Because plant HO enzymes are localized in the plastids (Muramoto et al., 1999; Gisk et al., 2010), a large amount of BV is expected to accumulate in these organelles. To determine whether smURFP could be used to visualize endogenous BV, we designed a construct to express smURFP fused with superfolder GFP (sfGFP) (Pédelacq et al., 2006; Fujii and Kodama, 2015) and a plastid-targeting peptide from the stromal protein ribulose bisphosphate carboxylase small chain 1A of *Arabidopsis* (*Arabidopsis thaliana*) (PT^sfGFP-smURFP) (Fig. 1A). sfGFP was used as a control fluorescent tag (Fig. 1A). To avoid interference from chlorophyll autofluorescence, we expressed

[1]Center for Bioscience Research and Education, Utsunomiya University, Tochigi 321-8505, Japan. [2]Graduate School of Regional Development and Creativity, Utsunomiya University, Tochigi 321-8505, Japan.
*These authors contributed equally to this work.

‡Author for correspondence (kodama@a.utsunomiya-u.ac.jp)

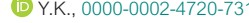 Y.K., 0000-0002-4720-7311

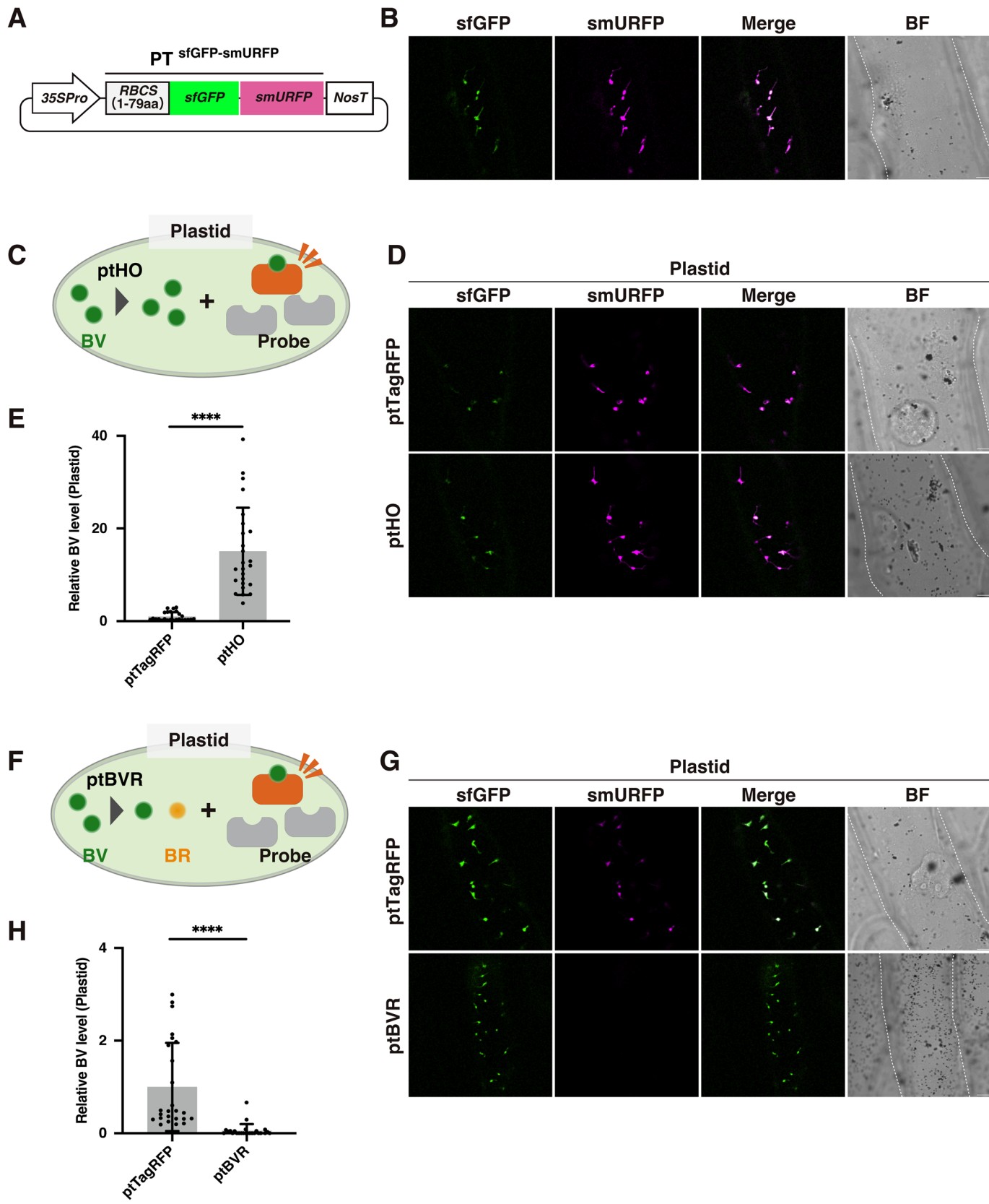

**Fig. 1.** See next page for legend.

the resulting constructs in epidermal cells of onion (*Allium cepa*) bulb scales, whose leucoplasts (plastids) do not contain chlorophyll pigments. When the PT$^{sfGFP-smURFP}$ construct was introduced into onion epidermal cells by biolistic bombardment, fluorescence of both sfGFP and smURFP was observed in the plastids (Fig. 1B), suggesting that they contained BV.

**Fig. 1. smURFP-based imaging of endogenous BV in plastids.**
(A) Illustration of a plasmid designed to express PT$^{sfGFP-smURFP}$. 35Spro, cauliflower mosaic virus 35S promoter; RBCS(1–79aa), a plastid-targeting peptide (N-terminal 79 amino acids of ribulose bisphosphate carboxylase small chain 1A from *Arabidopsis*); sfGFP, superfolder GFP; smURFP, small ultrared fluorescent protein; NosT, the terminator of *NOPALINE SYNTHASE* from *Rhizobium radiobacter* (*Agrobacterium tumefaciens*).
(B) Representative images of sfGFP and smURFP in plastids of onion cells expressing the PT$^{sfGFP-smURFP}$ probe. (C) Illustration of an experiment with PT$^{sfGFP-smURFP}$ and ptHO. (D) Representative images of sfGFP and smURFP in plastids of onion cells expressing the PT$^{sfGFP-smURFP}$ probe and ptHO. (E) Higher relative BV levels in plastids accumulating ptHO.
(F) Illustration of an experiment with PT$^{sfGFP-smURFP}$ and ptBVR.
(G) Representative images of sfGFP and smURFP in plastids of onion cells expressing the PT$^{sfGFP-smURFP}$ probe and ptBVR. (H) Lower relative BV levels in plastids accumulating ptBVR. (B,D,G) Scale bars: 20 µm. BF, bright field. Cell shapes are traced with dashed lines in BF images. (C,F) BR, bilirubin; BV, biliverdin; BVR, biliverdin reductase; HO, heme oxygenase; probe, sfGFP-smURFP. (E,H) BV levels were quantified using the fluorescence intensities of smURFP and sfGFP (see Materials and Methods). The mean BV level in plastids accumulating ptTagRFP was set to 1, and the relative BV level in plastids accumulating ptHO (E) or ptBVR (H) was determined. Fluorescence intensities from 25 plastids of five onion cells (five plastids each) were measured. Values are means±standard deviations (*n*=25). Asterisks indicate significant differences (Student's *t*-test, ****$P$<0.0001).

To confirm that the smURFP fluorescence originated from BV, we introduced the PT$^{sfGFP-smURFP}$ construct together with a gene encoding plastidic HO1 from *Arabidopsis* (ptHO, AT2G26670), which converts heme to BV (Fig. 1C). Plastid-targeted TagRFP-T (ptTagRFP) was used as a control. Cells expressing ptHO showed stronger smURFP fluorescence in plastids (Fig. 1D) and had significantly higher BV levels (estimated from fluorescence intensity) than cells expressing ptTagRFP (Fig. 1E). Conversely, BV levels were reduced when plastid-targeted BVR from rat (ptBVR) (Ishikawa et al., 2023; Ishikawa and Kodama, 2024), which converts BV to bilirubin, was co-expressed with PT$^{sfGFP-smURFP}$ (Fig. 1F–H). These results indicate that smURFP fluorescence can be used to visualize endogenous BV *in planta*.

## BV accumulation in peroxisomes
We next asked whether BV accumulates in peroxisomes despite the absence of HO in these organelles. When a gene encoding sfGFP-smURFP fused to a peroxisome localization signal (PS$^{sfGFP-smURFP}$) (Fig. 2A) was expressed in onion epidermal cells, smURFP fluorescence was observed in the peroxisomes (Fig. 2B). Note that in our preliminary experiments, we also detected smURFP fluorescence in the mitochondria but not in the ER (Fig. S1), suggesting that BV levels vary in different organelles. The ectopic expression of HO1 or BVR in peroxisomes (psHO or psBVR) led to greater or diminished smURFP fluorescence intensity (i.e. BV levels), respectively (Fig. 2C–H). Peroxisome-targeted TagRFP-T (psTagRFP) and Sirius (psSirius) were used as controls for psHO and psBVR, respectively (Fig. 2D,E,G,H). These results suggest that BV accumulates not only in plastids but also in peroxisomes. Because plant HO enzymes are not present in the peroxisomes, we hypothesized that BV was transported from the plastids to the peroxisomes.

## Bidirectional BV trafficking between plastids and peroxisomes
To test whether BV is trafficked from the plastids to the peroxisomes, we performed a trans-organelle assessment assay

(Ishikawa and Kodama, 2024); we co-expressed a peroxisome-targeted PS$^{sfGFP-smURFP}$ probe together with a gene encoding the plastid enzyme ptHO (Fig. 3A). In this way, we could increase BV biosynthesis in the plastids and monitor BV levels via PS$^{sfGFP-smURFP}$ in the peroxisomes (Fig. 3A). Cells with greater BV biosynthesis in the plastids also showed higher BV levels in the peroxisomes (Fig. 3B,C), indicating that BV was trafficked from the plastids to the peroxisomes.

To test whether BV is also trafficked from the peroxisomes to the plastids, we performed a similar trans-organelle assessment using the plastid-targeted PT$^{sfGFP-smURFP}$ probe and the peroxisome-targeted psHO enzyme (Fig. 3D). In this case, BV levels were higher in the peroxisomes when psHO was present; BV levels were monitored in the plastids via PT$^{sfGFP-smURFP}$ (Fig. 3D). Cells with higher BV levels in the peroxisomes also showed higher BV levels in the plastids (Fig. 3E,F), indicating that BV was trafficked from the peroxisomes to the plastids.

Taken together, our results provide strong evidence for bidirectional trafficking of BV between plastids and peroxisomes.

## BV trafficking from plastids to peroxisomes occurs constitutively under physiological conditions
Bidirectional trafficking of BV between plastids and peroxisomes was detectable when BV production was artificially increased (Fig. 3A–F). To determine whether BV trafficking occurs under normal physiological conditions, we performed trans-organelle assessments using BVR, which converts BV to bilirubin. When we introduced BVR into plastids, peroxisomal smURFP signals were significantly lower (Fig. S2). By contrast, when we introduced BVR into peroxisomes, smURFP signals in plastids were comparable with those of controls (Fig. S3). These results suggest that under physiological conditions, BV trafficking occurs constitutively from plastids to peroxisomes, but not from peroxisomes to plastids. The results obtained after psHO expression (Fig. 3D–F) suggest that BV trafficking from peroxisomes to plastids occurs only when high levels of BV are present in the peroxisomes.

## Bidirectional BV trafficking occurs via the cytosol
Bidirectional trafficking of BV between plastids and peroxisomes could occur by either direct or indirect transport (i.e. via the cytosol) of BV between the two organelles. To determine whether BV is exported to the cytosol from plastids and peroxisomes, we monitored BV levels using a cytosol-targeted sfGFP-smURFP probe (CS$^{sfGFP-smURFP}$) after enhancing BV levels in plastids or peroxisomes through expression of the corresponding HO enzyme (Fig. 4A,B). When BV levels were elevated in these organelles, BV levels in the cytosol were also higher in both cases (Fig. 4C,D). This observation indicates that BV is exported to the cytosol from both plastids and peroxisomes, suggesting that it passes through the cytosol as it moves between these organelles. To confirm this suggested pathway, we next asked whether BV is imported to the plastids and peroxisomes from the cytosol. When HO was expressed in the cytosol, BV levels were higher not only in the cytosol (Fig. S4) but also in the plastids and peroxisomes (Fig. 4E–J).

Taken together, these results indicate that inter-organelle trafficking of BV between the plastids and peroxisomes occurs through the cytosol (Fig. 5).

## DISCUSSION
We used smURFP-based imaging to examine the intracellular movement of BV, determining that it moves between the plastids

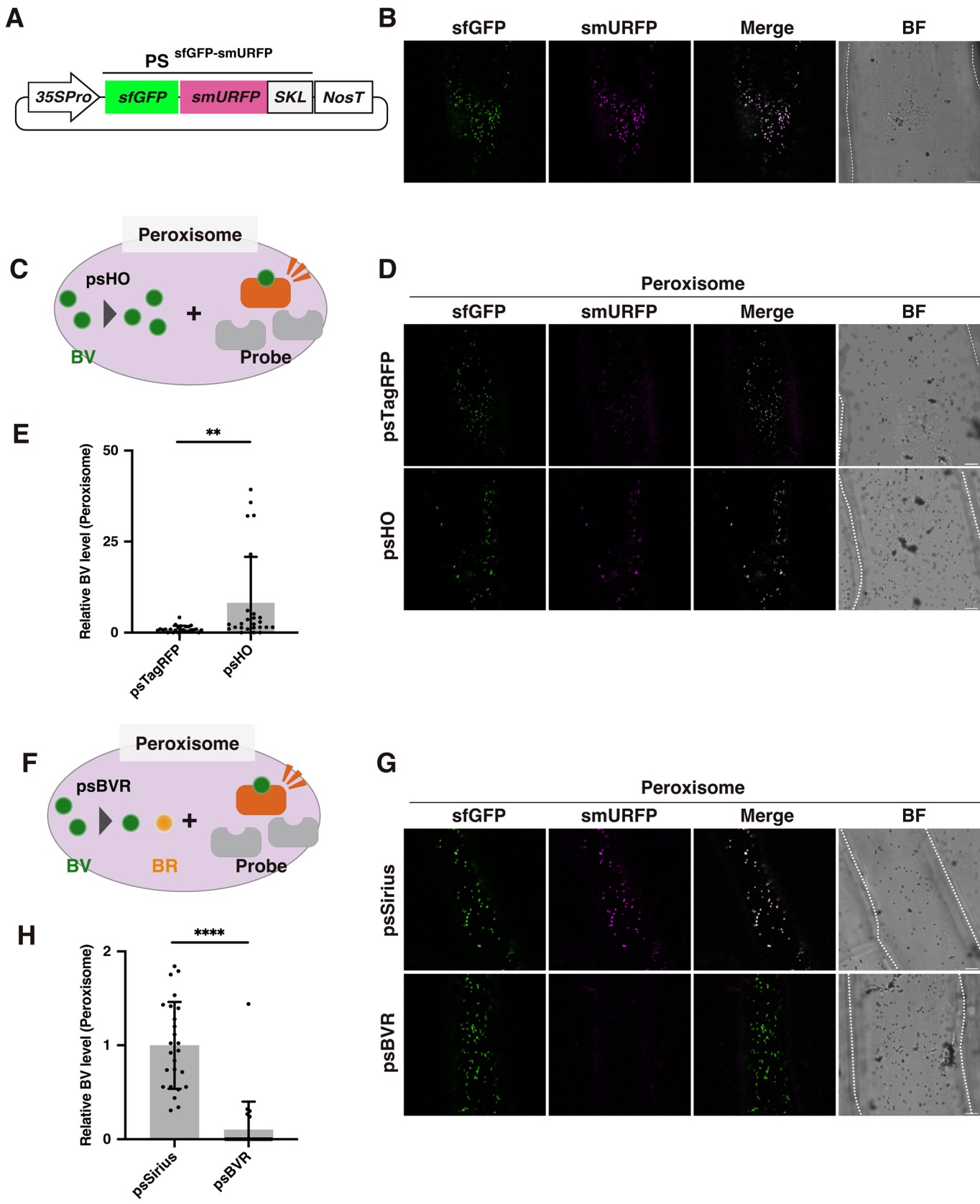

**Fig. 2.** See next page for legend.

and the peroxisomes through the cytosol. Our findings serve as a proof of concept for the visualization of metabolite trafficking at the organelle level.

In previous studies, we used UnaG technology to visualize endogenous bilirubin and reveal its subcellular distribution (Ishikawa et al., 2023; Ishikawa and Kodama, 2024). Bilirubin

**Fig. 2. smURFP-based imaging of endogenous BV in peroxisomes.**
(A) Illustration of a plasmid designed to express PS$^{sfGFP-smURFP}$. 35Spro, cauliflower mosaic virus 35S promoter; sfGFP, superfolder GFP; smURFP, small ultrared fluorescent protein; SKL, peroxisome-targeting peptide; NosT, the terminator of the *NOPALINE SYNTHASE* gene from *Rhizobium radiobacter* (*Agrobacterium tumefaciens*). (B) Representative images of sfGFP and smURFP in peroxisomes of onion cells expressing the PS$^{sfGFP-smURFP}$ probe. (C) Illustration of an experiment with PS$^{sfGFP-smURFP}$ and psHO. (D) Representative images of sfGFP and smURFP in peroxisomes of onion cells expressing the PS$^{sfGFP-smURFP}$ probe and psHO. (E) Higher relative BV levels in peroxisomes accumulating psHO. (F) Illustration of an experiment with PS$^{sfGFP-smURFP}$ and psBVR. (G) Representative images of sfGFP and smURFP in peroxisomes of onion cells expressing the PS$^{sfGFP-smURFP}$ probe and psBVR. (H) Lower relative BV levels in peroxisomes accumulating psBVR. (B,D,G) Scale bars: 20 μm. BF, bright field. Cell shapes are traced with dashed lines in BF images. (C,F) BR, bilirubin; BV, biliverdin; BVR, biliverdin reductase; HO, heme oxygenase; probe, sfGFP-smURFP. (E,H) BV levels were quantified using the fluorescence intensities of smURFP and sfGFP (see Materials and Methods). The mean BV level in peroxisomes accumulating psTagRFP (E) or psSirius (H) was set to 1, and the relative BV level in peroxisomes accumulating psHO (E) or psBVR (H) was determined. Fluorescence intensities from 25 peroxisomes of five onion cells (five peroxisomes each) were measured. Values are means±standard deviations (n=25). Asterisks indicate significant differences [Student's t-test, **P<0.01 (E) and ****P<0.0001 (H)].

was detected in the plastids, cytosol, ER, mitochondria, and peroxisomes, and enhanced bilirubin biosynthesis in plastids driven by ectopic expression of a gene for plastid-targeted BVR led to lower bilirubin levels in peroxisomes (Ishikawa and Kodama, 2024). We therefore hypothesized that the bilirubin precursor BV was trafficked from plastids to peroxisomes (Ishikawa and Kodama, 2024). In the present study, we confirmed this hypothesis by trans-organelle assessment using the BV-induced fluorescent protein smURFP and biosynthetic enzymes (HO and BVR). As expected, we observed BV trafficking from plastids to peroxisomes (Fig. 3A–C). We also observed BV trafficking in the opposite direction, from peroxisomes to plastids, when BV production was elevated (Fig. 3D–F), and demonstrated that BV passes through the cytosol (Fig. 4). Trans-organelle assessment using smURFP and biosynthetic enzymes is a powerful experimental approach for investigating BV behavior at the subcellular level. In our preliminary experiments, we also detected smURFP fluorescence in the mitochondria, but not in the ER (Fig. S1). Given that bilirubin was previously detected in both mitochondria and the ER (Ishikawa and Kodama, 2024), BV abundance in the ER appears to be below the detection limit of our smURFP-based imaging. Nevertheless, BV trafficking to and from these organelles remains important and can be effectively explored using trans-organelle assessment.

Based on our data using trans-organelle assessment, BV trafficking from plastids to peroxisomes appears to be a constitutive process, whereas trafficking from peroxisomes to plastids is conditional, occurring in response to an increase in BV levels in peroxisomes (Fig. 3D–F; Fig. S3). As in the case of plastids (Ishikawa et al., 2023), BV in peroxisomes appears to play a role in redox homeostasis through its conversion to bilirubin (a strong antioxidant) and/or its own antioxidant activity. We expect that BV trafficking from peroxisomes to plastids serves as a feedback mechanism, indicating that a sufficient abundance of BV has been achieved in the peroxisomes for the BV synthetic pathway in plastids.

smURFP is known to bind to phycocyanobilin as well as BV as a fluorophore (Rodriguez et al., 2016). Land plants do not produce phycocyanobilin but instead biosynthesize PΦB, which has a similar molecular structure. Given this structural similarity,

smURFP might also bind to PΦB. Although we confirmed BV-induced smURFP fluorescence using related enzymes (HO and BVR) (e.g. Figs 1,2), it is possible that the current smURFP-based imaging technique could detect PΦB as well as BV in plant cells. However, in dark-grown *Arabidopsis hy2-1* mutants defective in PΦB synthase (Kohchi et al., 2001), smURFP fluorescence was increased in the plastid, compared to wild-type cells (Fig. S5), suggesting that smURFP does not bind to PΦB. Alternatively, in wild-type cells, PΦB may be rapidly exported from the plastid to the cytosol (Fig. S5); therefore, further studies are needed to clarify the specificity. Nonetheless, we consider it unlikely that PΦB was detected by smURFP in epidermal cells of onion bulb scales in the present study. There are two reasons for this view. First, phytochrome is unlikely to function in these epidermal cells, as they are located inside the bulb and are not exposed to light. Under such conditions, phytochrome activity would be dispensable, and PΦB biosynthesis would likely be strongly suppressed. Second, PΦB synthase may not function properly in these cells. The reduction of BV by PΦB synthase to form PΦB in the plastids requires reduced ferredoxin as an electron donor; this is supplied by ferredoxin–NADP$^+$ reductase, which transfers electrons from NADPH to ferredoxin (Kohchi et al., 2001). In preliminary experiments, we examined whether *Arabidopsis* PΦB synthase (HY2, AT3G09150) could function in onion cells. HY2 was expressed in the plastids, and smURFP fluorescence was detected in the peroxisomes (Fig. S6). If HY2 had been active in the plastids, smURFP fluorescence in the peroxisomes would have been lower, as observed upon plastid expression of ptBVR (Fig. S2). However, although we expressed HY2 and compatible ferredoxin 2 (AT1G60950) (Chiu et al., 2010), we did not observe a difference in peroxisomal fluorescence levels (Fig. S6), suggesting that the HY2 enzyme was not functional in the onion plastids. Because these plastids are leucoplasts, which lack photosynthetic activity, NADPH derived from photosystem I is not produced. Although leucoplasts can generate some NADPH through metabolic pathways such as the oxidative pentose phosphate pathway (Ernes and Neuhaus, 1997), available reducing power and possibly ferredoxin-dependent electron transfer may be insufficient to support HY2 activity in onion epidermal cells. We therefore believe that PΦB is not biosynthesized in the plastids of onion-bulb-scale epidermal cells and that only BV was detected with smURFP in these cells.

Co-expression of the sfGFP-smURFP probe with the HO enzyme in the cytosol or peroxisomes resulted in higher BV levels (Fig. 2C–E; Fig. S4). This result implied the presence of heme, the precursor of BV, in these cellular compartments (Tenhunen et al., 1968, 1969). To date, the presence of heme has been inferred from the subcellular localizations of heme-binding proteins (e.g. cytochrome c) and ferrochelatase, which catalyzes the terminal step of heme biosynthesis (Matringe et al., 1994). Although genetically encoded probes have been developed for live-cell imaging of heme, they require the use of ratiometric or fluorescence resonance energy transfer microscopy techniques (Hanna et al., 2016; Song et al., 2015). By contrast, our experimental system, which combines a smURFP probe with an HO enzyme, reveals the presence of heme in various subcellular compartments through simple observation of increased smURFP fluorescence.

In *Arabidopsis*, four HOs are known to localize to the plastids (Muramoto et al., 1999; Gisk et al., 2010). A recent study demonstrated that HO1 (AT2G26670) localizes to both the plastids and the cytosol owing to the presence of two alternative transcription start sites (Chen et al., 2024). This finding suggests that BV is biosynthesized from heme by HO in both the plastids

Biology Open

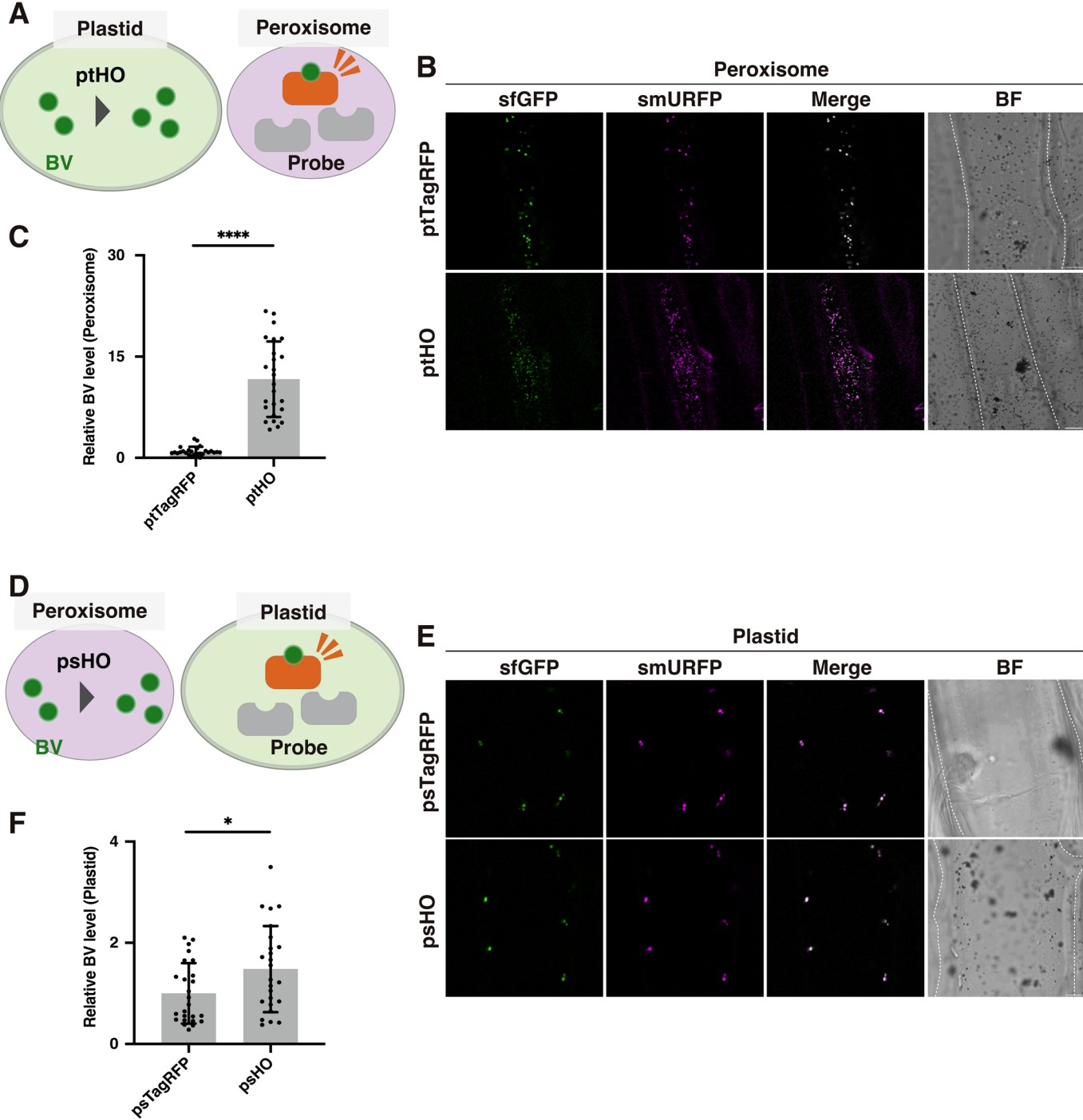

**Fig. 3. Bidirectional BV trafficking between plastids and peroxisomes.** (A) Illustration of trans-organelle assessment with PS$^{sfGFP\text{-}smURFP}$ in peroxisomes and ptHO in plastids. (B) Representative images of sfGFP and smURFP in peroxisomes of onion cells expressing the PS$^{sfGFP\text{-}smURFP}$ probe and ptHO. (C) Higher relative BV levels in peroxisomes of cells expressing ptHO. (D) Illustration of trans-organelle assessment with PT$^{sfGFP\text{-}smURFP}$ in plastids and psHO in peroxisomes. (E) Representative images of sfGFP and smURFP in plastids of onion cells expressing the PT$^{sfGFP\text{-}smURFP}$ probe and psHO. (F) Higher relative BV levels in plastids of cells expressing psHO. (A,D) BV, biliverdin; HO, heme oxygenase; probe, sfGFP-smURFP. (B,E) Scale bars: 20 μm. BF, bright field. Cell shapes are traced with dashed lines in BF images. (C,F) BV levels were quantified using the fluorescence intensities of smURFP and sfGFP (see Materials and Methods). The mean BV level in cells expressing psTagRFP was set to 1, and the relative BV level in cells expressing ptHO (C) or psHO (F) was determined. Fluorescence intensities from 25 peroxisomes (C) or plastids (F) of five onion cells (five peroxisomes or plastids each) were measured. Values are means±standard deviations ($n$=25). Asterisks indicate significant differences [Student's $t$-test, ****$P<0.0001$ (C) and *$P<0.05$ (F)].

and the cytosol. In the present study, we revealed that BV is present not only in the plastids and cytosol but also in the peroxisomes (Fig. 5). Whereas BV levels in the plastids and cytosol are regulated by the biosynthetic capacities of these compartments, BV levels in the peroxisomes are dependent on import from the plastids and cytosol. This hierarchical organization of BV distribution highlights the regulatory relationships between subcellular compartments with and without BV biosynthetic capacity.

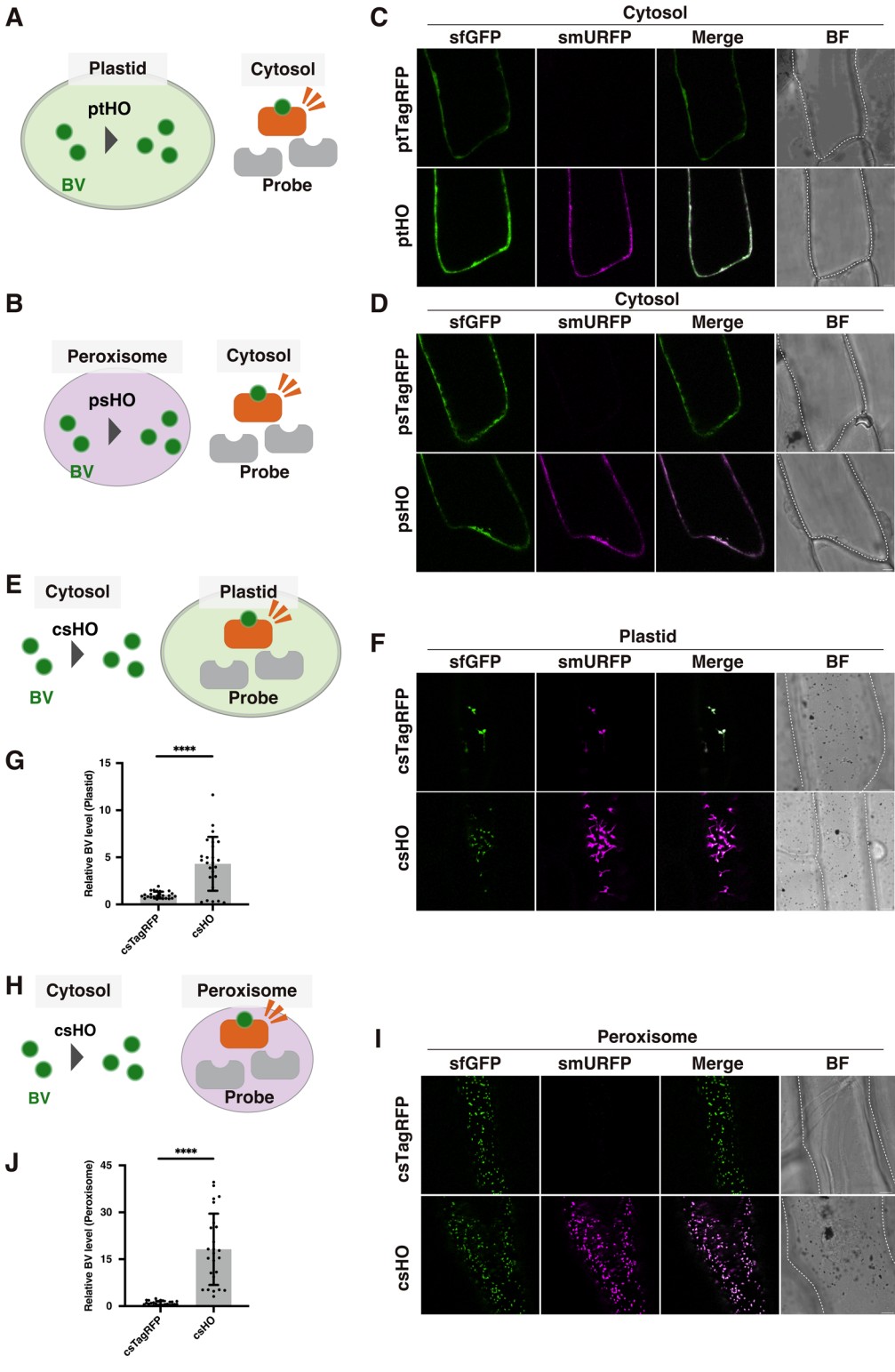

**Fig. 4. Bidirectional BV trafficking via the cytosol.** (A,B) Illustration of trans-organelle assessment using CS$^{sfGFP-smURFP}$ in the cytosol with ptHO in plastids (A) or psHO in peroxisomes (B). (C,D) Representative images of sfGFP and smURFP in the cytosol of onion cells expressing the CS$^{sfGFP-smURFP}$ probe with ptHO (C) or psHO (D). (E) Illustration of trans-organelle assessment using PT$^{sfGFP-smURFP}$ in plastids with csHO in the cytosol. (F) Representative images of sfGFP and smURFP in plastids of onion cells expressing the PT$^{sfGFP-smURFP}$ probes with csHO. (G) Higher relative BV levels in plastids of cells expressing csHO. (H) Illustration of trans-organelle assessment using PS$^{sfGFP-smURFP}$ in peroxisomes with csHO in the cytosol. (I) Representative images of sfGFP and smURFP in plastids of onion cells expressing the PS$^{sfGFP-smURFP}$ probes with csHO. (J) Higher relative BV levels in peroxisomes of cells expressing csHO. (A,B,E,H) BV, biliverdin; HO, heme oxygenase; probe, sfGFP-smURFP. (C,D,F,I) Scale bars: 20 µm. BF, bright field. Cell shapes are traced with dashed lines in BF images. (G,J) BV levels were quantified using the fluorescence intensities of smURFP and sfGFP (see Materials and Methods). The mean BV level in cells expressing csTagRFP was set to 1, and the relative BV level in cells expressing csHO was determined. Fluorescence intensities from 25 plastids (G) or peroxisomes (J) of five onion cells (five peroxisomes or plastids each) were measured. Values are means±standard deviations ($n$=25). Asterisks indicate significant differences (Student's $t$-test, ****$P$<0.0001).

We previously demonstrated that bilirubin is produced by a nonenzymatic reaction between BV and NADPH in the plastids but cannot move from the plastids to other cellular compartments (i.e. the peroxisomes, cytosol, mitochondria, or ER) (Ishikawa and Kodama, 2024). Nonetheless, we detected the presence of bilirubin in these non-plastid compartments (Ishikawa and Kodama, 2024). In the present study, we demonstrated that BV, which is biosynthesized by HO in the plastids, can move to the cytosol and peroxisomes (Fig. 5).

On the basis of these observations, we suggest that BV is a mobile metabolite within the cell and that bilirubin is biosynthesized by a reaction between BV and NADPH in various cellular compartments after delivery of BV from the plastids (Fig. 5). Because BV has a lower antioxidant activity than bilirubin (Jansen et al., 2010), it is less likely to affect cellular metabolism during the transport process. Some intermediate metabolites with low physiological activity may play roles in facilitating its movement between cellular compartments.

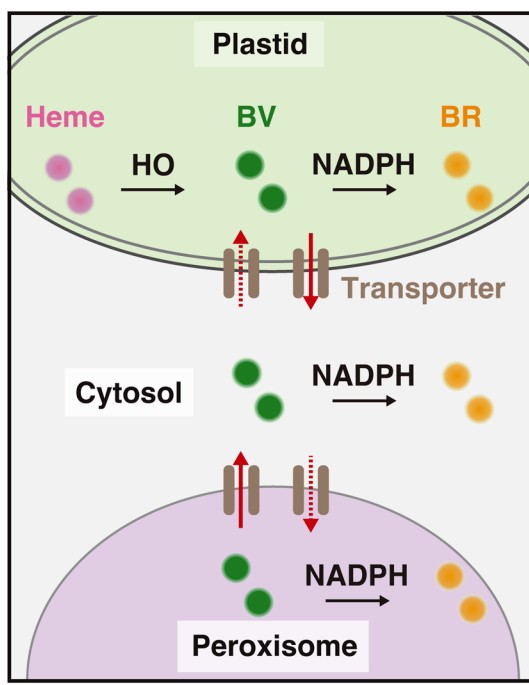

**Fig. 5. A model of BV trafficking and bilirubin distribution.** BR, bilirubin; BV, biliverdin; HO, heme oxygenase; NADPH, nicotinamide adenine dinucleotide phosphate.

Because BV is hydrophilic, it is impermeable to biological membranes. Nevertheless, exogenous BV has been shown to be transported into both animal cells (HeLa cells) and plant cells (*Arabidopsis*) (Shemetov et al., 2017; Parks and Quail, 1991) via unknown pathways, including specific transporters and/or endocytosis. Consistent with this, although BV moves among the plastids, peroxisomes, and cytosol (Fig. 5), its hydrophilicity prevents its passive diffusion across organelle membranes; instead, its membrane transport appears to require specific transporter proteins. In human cells, the ATP-binding cassette (ABC) transporter ABCB10 on the inner mitochondrial membrane (Shum et al., 2021) exports BV from the mitochondria to the cytosol (Shum et al., 2021). Given the discovery of ABCB10 in animal cells, we speculate that specific ABC transporters may also transport BV between organelles in plant cells (Fig. 5), but this possibility remains to be tested. Compared with the human genome (48 genes) (Dean et al., 2001), plant genomes contain many more ABC transporter genes: 132 in *Arabidopsis*, 261 in soybean (*Glycine max*), and 128 in rice (*Oryza sativa*) (Verrier et al., 2008; Mishra et al., 2019; Schulz and Kolukisaoglu, 2006). When subcellular localizations and topologies of the 132 *Arabidopsis* ABC transporters were predicted *in silico*, most proteins were predicted to localize to the plasma membrane, but some proteins were predicted to localize to organelles, including plastids and peroxisomes (Table S1). It will be necessary to develop a screening strategy for the identification of potential BV transporters in various plant organelles.

Very little is known about the intracellular distribution and inter-organelle trafficking of metabolites, owing to technical limitations. The present study describes a technique for analyzing the intracellular locations and movements of metabolites by combining a fluorescent probe (i.e. smURFP) with biosynthetic enzymes (i.e. HO and BVR). Similar experimental designs using other fluorescent probes and related enzymes could be used to clarify the behaviors of various metabolites, revealing where and how they are located and trafficked within the cell.

## MATERIALS AND METHODS
### Plasmid construction for sfGFP-smURFP probes

All primers used are listed in Table S2. To construct the plasmid encoding PT$^{sfGFP-smURFP}$ (the plastid probe), we designed plastid-targeted sfGFP-smURFP, composed of a plastid-targeting peptide [the N-terminal 79 amino acids of ribulose bisphosphate carboxylase small chain 1A (RBCS) from *Arabidopsis*] (Osaki and Kodama, 2017), sfGFP (Pédelacq et al., 2006; Fujii and Kodama, 2015), and smURFP (Rodriguez et al., 2016). DNA encoding PT$^{sfGFP-smURFP}$ was synthesized as a synthetic gene (Integrated DNA Technologies) (Fig. S7). With this as a template, DNA fragments were amplified by PCR with primers #1 and #2 (Table S2) and cloned into the pDONR207 vector by the BP reaction using Gateway cloning technology (Thermo Fisher Scientific) to generate pDONR207-RBCS(1–79)-sfGFP-smURFP.

To construct the plasmid encoding CS$^{sfGFP-smURFP}$ (the cytosol probe), DNA fragments were amplified by PCR with primers #3 and #2 and the template pDONR207-RBCS(1–79)-sfGFP-smURFP. The resulting fragment was cloned into the pDONR207 vector via the BP reaction to obtain pDONR207-sfGFP-smURFP.

To construct the plasmid encoding PS$^{sfGFP-smURFP}$ (the peroxisome probe), the tripeptide SKL was added to the C terminus of sfGFP-smURFP as a peroxisome localization signal (Gould et al., 1987, 1989). To add the peroxisome localization signal, DNA fragments were amplified by PCR with primers #3 and #4 and the template pDONR207-sfGFP-smURFP. The resulting fragment was cloned into the pDONR207 vector by the BP reaction to obtain pDONR207-sfGFP-smURFP-SKL.

To construct the plasmid encoding MT$^{sfGFP-smURFP}$ (the mitochondrial probe), DNA fragment 1 containing the mitochondrial targeting signal (1–55 aa) of *Arabidopsis* cysteine synthase (AtCYSC1, AT3G61440) (Yamaguchi et al., 2000) was amplified by PCR with primers #5 and #6 and an *Arabidopsis* cDNA library as a template. DNA fragment 2 comprising sfGFP-smURFP was amplified by PCR with primers #7 and #2 and the template pDONR207-sfGFP-smURFP. Fragments 1 and 2 were fused by recombinant PCR to generate AtCYSC1(1–55)-sfGFP-smURFP. The resulting fragment was cloned into the pDONR207 vector by the BP reaction to obtain pDONR207-AtCYSC1(1–55)-sfGFP-smURFP.

To construct the plasmid encoding ER$^{sfGFP-smURFP}$ (the ER probe), DNA fragment 3 was amplified by PCR with primers #8 and #9 and the template pENTR1A-SP-mCherry-UnaG-HDEL (Ishikawa and Kodama, 2024). DNA fragment 4 was amplified by PCR with primers #10 and #11 and the template pDONR207-sfGFP-smURFP. Fragments 3 and 4 were fused by recombinant PCR to generate *SP-sfGFP-smURFP-HDEL*. The resulting fragment was cloned into the pDONR207 vector by the BP reaction to obtain pDONR207-SP-sfGFP-smURFP-HDEL.

pDONR207-RBCS(1–79)-sfGFP-smURFP, pDONR207-sfGFP-smURFP, pDONR207-sfGFP-smURFP-SKL, pDONR207-AtCYSC1(1–55)-sfGFP-smURFP, or pDONR207-SP-sfGFP-smURFP-HDEL was mixed with pGWT35S (Addgene plasmid #182790) (Fujii et al., 2018), and the Gateway LR reaction (Thermo Fisher Scientific) was performed to obtain pGWT35S-RBCS(1–79)-sfGFP-smURFP, pGWT35S-sfGFP-smURFP, pGWT35S-sfGFP-smURFP-SKL, pGWT35S-AtCYSC1(1–55)-sfGFP-smURFP, or pGWT35S-SP-sfGFP-smURFP-HDEL. The pGWT35S vectors constructed are listed in Table S3.

### Plasmid construction for the enzyme series

All primers used are listed in Table S2. TagRFP-T (Shaner et al., 2008) and Sirius (Tomosugi et al., 2009) were used as control proteins. DNA fragments for csTagRFP (cytosol type) and psTagRFP (peroxisome type) were amplified by PCR with the template pDONR207-RBCS1a(1–79)-TagRFP-T (Osaki and Kodama, 2017). Primers #12 and #13 and primers #12 and #14 were used for csTagRFP and psTagRFP, respectively. DNA fragments for psSirius (peroxisome type) were amplified by PCR using the Sirius/pRSETB plasmid (Addgene plasmid #51956) as a template with primers #3 and #15 to obtain *Sirius-SKL*. For ptSirius (plastid type), two DNA fragments encoding the plastid-targeting peptide [RBCS1a(1–79)] and Sirius were amplified by PCR. The *RBCS1a(1–79)* fragment was amplified from pGWB602-RBCS-tagRFP (Addgene plasmid #213868) (Ichikawa et al., 2024) using primers #1 and #16, whereas the *Sirius*

fragment was amplified from the Sirius/pRSETB using primers #17 and #18. These two fragments were fused by recombinant PCR using primers #1 and #18 to obtain *RBCS1(1–79)-Sirius*. The resulting DNA fragments were cloned into pDONR207 by the BP reaction to obtain pDONR207-TagRFP-T, pDONR207-TagRFP-T-SKL, pDONR207-Sirius-SKL, and pDONR207-RBCS(1–79)-Sirius.

For the plastid-targeted HO-TagRFP fusion protein (ptHO), the DNA fragment of HO1 (AT2G26670) was amplified by PCR using an *Arabidopsis* cDNA library with primers #19 and #20. The DNA fragment for TagRFP-T was also amplified using pDONR207-RBCS1a(1–79)-TagRFP-T (Osaki and Kodama, 2017) as a template with primers #21 and #13. The DNA fragment for HO1 was then fused to the DNA fragment for TagRFP-T by recombinant PCR to obtain *HO1(1–282)-TagRFP-T*, which was cloned into the pDONR207 vector by the BP reaction to obtain pDONR207-HO1(1–282)-TagRFP-T. For the cytosol-targeted HO-TagRFP fusion protein (csHO), the DNA fragment for HO1 without the plastid-targeting peptide (1–54 aa) (Muramoto et al., 1999) was amplified by PCR using pDONR207-HO1(1–282)-TagRFP-T as a template with primers #22 and #13 and then cloned into pDONR207 by the BP reaction to obtain pDONR207-HO1(55–282)-TagRFP-T. For the peroxisome-targeted HO-TagRFP fusion protein (psHO), the DNA fragment for HO1 without the plastid-targeting peptide (1–54 aa) was amplified by PCR using pDONR207-HO1(1–282)-TagRFP-T as a template with primers #22 and #14 and then cloned into pDONR207 by the BP reaction to obtain pDONR207-HO1(55–282)-TagRFP-T-SKL.

To produce a vector for plastid-targeted BVRA from rat (ptBVR), pENTR1A-RBCS1a(1–79)-BVRA-flag (previously designated pENTR1A-TP-BVRA) (Ishikawa and Kodama, 2024) was used. To produce a peroxisome-targeted BVRA (psBVR), a DNA fragment for BVRA-flag was amplified by PCR using pENTR1A-RBCS(1–79)-BVRA-flag with primers #23 and #24 and then cloned into pENTR1A by In-Fusion Cloning (Clontech) to obtain pENTR1A-BVRA-flag-SKL.

To produce a vector for plastid-targeted HY2 (PΦB synthase) (ptHY2) fused to Sirius, a DNA fragment for HY2 (AT3G09150) was amplified by PCR using an *Arabidopsis* cDNA library with primers #25 and #26. DNA fragments for Sirius were also amplified using the Sirius/pRSETB as a template with primers #27 and #28. The HY2 and Sirius fragments were fused by recombinant PCR to obtain *HY2-Sirius*, which was then cloned into the pDONR/Zeo vector to obtain pDONR/Zeo-HY2-Sirius.

To produce a vector for plastid-targeted ferredoxin (Fd) (Chiu et al., 2010) fused to TagRFP-T, a DNA fragment for Fd2 (AT1G60950) was amplified by PCR using an *Arabidopsis* cDNA library with primers #29 and #30. DNA fragments for TagRFP-T were also amplified using pDONR207-TagRFP-T as a template with primers #31 and #13. The Fd2 and TagRFP-T fragments were fused by recombinant PCR to obtain Fd2-TagRFP-T, which was then inserted into pDONR207 by the BP reaction to obtain pDONR207-Fd2-TagRFP-T.

pDONR207-TagRFP-T, pDONR207-TagRFP-T-SKL, pDONR207-Sirius-SKL, pDONR207-RBCS(1–79)-Sirius, pDONR207-HO1(1–282)-TagRFP-T, pDONR207-HO1(55–282)-TagRFP-T, pDONR207-HO1(55–282)-TagRFP-T-SKL, pENTR1A-RBCS1a(1–79)-BVRA-flag, pENTR1A-BVRA-flag-SKL, pDONR/Zeo-HY2-Sirius, or pDONR207-Fd2-TagRFP-T was mixed with pGWT35S (Fujii et al., 2018), and an LR reaction was performed to generate pGWT35S-TagRFP-T, pGWT35S-TagRFP-T-SKL, pGWT35S-Sirius-SKL, pGWT35S-RBCS(1–79)-Sirius, pGWT35S-HO1(1–282)-TagRFP-T, pGWT35S-HO1(55–282)-TagRFP-T, pGWT35S-HO1(55–282)-TagRFP-T-SKL, pGWT35S-RBCS(1–79)-BVRA-flag, pGWT35S-BVRA-flag-SKL, pGWT35S-HY2-Sirius, or pGWT35S-Fd2-TagRFP-T. The pGWT35S vectors constructed are listed in Table S3.

## Particle bombardment

We introduced constructs into epidermal cells of onion (*A. cepa*) bulb scales by particle bombardment using the PDS-1000/He biolistic transformation system (Bio-Rad) (Osaki and Kodama, 2017) with a helium pressure of 1100 psi and a target distance of 60 mm. After bombardment, the cells were incubated in darkness for 24 h, and fluorescence was observed by confocal laser scanning microscopy.

## Confocal laser scanning microscopy

Fluorescence was observed in onion epidermal cells using an SP8X confocal laser scanning microscope system equipped with a hybrid detector (HyD) (Leica Microsystems), using a white light laser (WLL) for excitation. Images were acquired at 512×512 pixels, with HyD detection mode set to counting and a scan speed of 100 Hz. sfGFP was excited with a 488 nm laser obtained from the WLL and detected using the HyD in the wavelength range of 500–530 nm. smURFP was excited with a 633 nm laser from the WLL and detected using the HyD in the wavelength range of 660–690 nm. Images acquired by confocal laser scanning microscopy were exported as TIFF files. All fluorescence images in grayscale are included in Figs S8 and S9.

## Quantification of BV levels

The fluorescence intensities of smURFP and sfGFP were measured from the TIFF files using ImageJ Fiji (Schindelin et al., 2012). According to the bilirubin quantification method (Ishikawa et al., 2023), BV levels were calculated using the following formula:

$$\text{BV level} = \frac{1}{n}\sum_{i=1}^{i}\left(x_i\frac{\bar{y}}{y_i}\right),$$

where $x$ represents the smURFP measurement value, $y$ represents the sfGFP measurement value (Ishikawa et al., 2023), and $\bar{y}$ represents the mean sfGFP intensity. Fluorescence intensities from 25 regions of five onion cells (five regions each) were measured. The experiments were repeated at least three times to confirm reproducibility (Figs S10 and S11). In each experiment, one to three onion bulbs were used.

## Acknowledgements

We thank Dr. Hiroki Ikeda (Utsunomiya University) for providing *A. cepa*; Dr. Takeharu Nagai (Osaka University) for providing the Sirius/pRSETB plasmid (Addgene plasmid #51956); Drs Kazuya Ishikawa (Okayama University) and Shintaro Ichikawa (Utsunomiya University) for technical assistance with plasmid construction; and Dr. Chonprakun Thagun (Utsunomiya University) for valuable discussion.

## Competing interests

The authors declare no competing or financial interests.

## Author contributions

Conceptualization: Y.K.; Data curation: M.S., K.Y.; Formal analysis: M.S., K.Y.; Funding acquisition: Y.K.; Investigation: M.S., K.Y., H.T.; Methodology: Y.K.; Resources: Y.K.; Supervision: Y.K.; Visualization: M.S., K.Y., H.T., Y.K.; Writing – original draft: Y.K.; Writing – review & editing: M.S., K.Y., H.T., Y.K.

## Funding

This work was supported by the Japan Society for the Promotion of Science (JSPS) KAKENHI (grant numbers 24K01716 and 25K21730); the Ministry of Education, Culture, Sports, Science and Technology (MEXT) KAKENHI (grant number 25H01340); and the Shorai Foundation for Science and Technology. Open Access funding provided by the Shorai Foundation for Science and Technology. Deposited in PMC for immediate release.

## Data and resource availability

All relevant data and details of resources can be found within the article and its supplementary information.

## Declaration of generative AI and AI-assisted technologies in the writing process

The authors used ChatGPT to improve English grammar during the preparation of this manuscript.

## Peer review history

The peer review history is available online at https://journals.biologists.com/bio/lookup/doi/10.1242/bio.062435.reviewer-comments.pdf

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
