## [Peer Review File · Biology Open]

Live-cell tracking of biliverdin trafficking reveals metabolic exchange between plastids and peroxisomes

Mone Shibata, Keiji Yoshida, Hitomi Takahashi and Yutaka Kodama
10.1242/bio.062435

Editor: Marie Monniaux

Review timeline

Original submission:	18 December 2025
Editorial decision:	7 January 2026
First revision received:	3 April 2026
Accepted:	20 April 2026

Original submission

First decision letter

MS ID#: bio.062435

MS Title: Live-cell tracking of biliverdin trafficking reveals metabolic exchange between plastids and peroxisomes

Authors: Mone Shibata; Keiji Yoshida; Hitomi Takahashi; Yutaka Kodama

I have now reached a decision on the above manuscript.

The reviewer reports are shown at the bottom of this email.

As you will see, the reviewers gave favourable reports, but raised some critical points that will require amendments to your manuscript. I hope that you will be able to carry these out, because we would like to be able to accept your paper. Please provide a point-by-point answer to all reviewers' comments.

At this stage, we also ask you to ensure your manuscript complies with our formatting guidelines - please see our manuscript preparation guidelines for details. Provided you are able to fully address the referees' comments, we are positive about publication of your paper (we accept over 95% of revision submissions) and therefore hope you won't mind any extra work involved in reformatting your manuscript at this point.

Please upload both a 'clean' version of your Word file, along with a highlighted version clearly showing where you have made changes in the revised manuscript. Please avoid using 'Track changes' in Word files as these are lost in PDF conversion.

I should be grateful if you would also provide a point-by-point response detailing how you have dealt with the points raised by the reviewers in the 'Response to Reviewers' box. Please attend to all of the reviewers' comments. If you do not agree with any of their criticisms or suggestions please explain clearly why this is so.

Reviewer 1

Comments for the author

In this study the authors examine the intracellular distribution of biliverdin (BV) in plant cells by selectively targeting a fluorescent protein that requires BV to fluoresce to different organelles. Through this elegant technique, the authors convincingly demonstrate the presence of BV in plant cells, and the ability of BV synthesized in some organelles to be transported to others via the cytosol. The demonstration that selective targeting of enzymes that increase or decrease the levels of BV to specific organelles can influence smURFP fluorescence in other organelles is strong evidence for their model of inter-organelle BV transport. Importantly, the authors also adequately address potential complications/alternative explanations of using smURFP as a BV sensor in the discussion. I have a few suggestions that I believe would help improve the clarity and reproducibility of this manuscript.

1. Most figure legends mentions a "bright field" image, but I do not see one in any of the micrographs. Including BF images would help orient readers who are unfamiliar with the system being used.
2. Most figures include the number of cells analyzed but do not mention the number of true biological replicates. This information is necessary to understand the reproducibility of the findings.
3. The magenta panels in most figures are difficult to see. While not necessary to change, it would be better to use grayscale images of the individual fluorescent channels and color images for the merged panels.
4. The use of psSirius is mentioned in figure legends but not the results section.
5. There are a few duplicated references (Ishikawa and Kodama, Kochi et al.,)
6. Figure S4 is only mentioned in the discussion
7. In some of the figures it becomes difficult to track to which organelles the smURFP probe and the various enzymes are being targeted. It would help to add additional labels to the micrographs.
8. Using "relative BV level" as the y-axis label in several of the graphs (1E,H ; 2E,H and so on) is a little misleading. The actual measurement is the normalized smURFP fluorescence, which is used as a proxy for the levels of BV
9. The discussion and model figure could benefit from a further examination of the transport of BV. The model could use additional explanatory text in the figure legend. Do the red arrows indicate active transport? The authors reference the active transport via ABC transporters identified by Shum et al., 2021, but exogenous BV added to cultured cells has been shown to enhance smURFP fluorescence, implying that BV must be able to cross the PM of some cells via diffusion or import (Shemetov et al., 2017). Could some of these findings be explained by certain organelles membranes are more/less BV-permeable due to lipid as well as protein content?

Reviewer 2

Comments for the author

This manuscript reports a great imaging-based strategy to visualize inter-organelle metabolite trafficking in plant cells using a smURFP (small ultra-red fluorescent protein)-based biliverdin (BV) sensor. By combining compartment-targeted smURFP with enzymatic manipulation of BV biosynthesis and degradation, the authors establish a "trans-organelle assessment" framework that enables dynamic inference of metabolite movement between plastids and peroxisomes. The work provides a technically sound proof of concept for probing organelle-level metabolite exchange and represents a meaningful methodological advance. However, some conceptual and interpretative

issues require clarification to strengthen the physiological relevance and specificity of the conclusions.

Major comments

(1) Innovation and experimental design

The purposing of smURFP as a genetically encoded reporter for endogenous BV in plant cells is technically creative and well executed. The use of onion epidermal cells to minimize chlorophyll autofluorescence is appropriate and strengthens signal interpretation. The "trans-organelle assessment" assay wherein BV levels are artificially elevated in one organelle through heme oxygenase (HO) expression and monitored in distal compartments, which constitutes a robust experimental design that goes beyond static localization. This approach convincingly demonstrates inter-organelle metabolite mobility and could be broadly applicable to other metabolic systems.

(2) Bidirectional versus physiological trafficking

The data indicate that BV trafficking between plastids and peroxisomes is bidirectional under conditions of artificially elevated BV, whereas under basal conditions the dominant direction appears to be plastid-to-peroxisome. This distinction is important but currently underdeveloped. The authors propose that peroxisome-to-plastid trafficking occurs only when BV levels exceed a certain threshold. Can the authors discuss plausible physiological or stress-related scenarios (e.g., oxidative stress, heme overload, altered tetrapyrrole flux) that might generate such elevated BV concentrations in peroxisomes *in vivo*?

(3) Specificity of smURFP for BV versus phytochromobilin

The manuscript appropriately acknowledges that smURFP may also bind phytochromobilin (PΦB), given its structural similarity to known ligands. The authors argue that PΦB is unlikely to accumulate in dark-grown onion bulb scales and further support this claim with negative results from HY2 and ferredoxin co-expression. While these controls are informative, have the authors considered validating probe specificity in a tissue or genetic background where PΦB accumulation is expected (e.g., light-grown tissues or HY2-overexpressing systems)? Such validation would more definitively establish the extent of smURFP cross-reactivity in planta.

(4) Speculation on BV transport mechanisms

The authors speculate that ATP-binding cassette (ABC) transporters may mediate BV translocation across organellar membranes, drawing analogy to human ABCB10. While this hypothesis is reasonable given the hydrophilic nature of BV, the discussion remains overly general. We suggest authors could strengthen this section by briefly highlighting which Arabidopsis ABC transporter subfamilies or members (based on predicted subcellular localization or known tetrapyrrole-associated functions) represent the most plausible candidates for future investigation.

Minor Comments

(1) Reference repeats

There are duplicate entries for single references, such as Kohchi et al. (2001a, b) and Ishikawa and Kodama (2024a, b). In addition, some references contain incomplete titles (e.g., Tenhunen et al., 1969).

(2) Mitochondrial versus ER localization

Preliminary observations suggest BV accumulation in mitochondria but not in the endoplasmic reticulum (ER). Given previous reports of bilirubin localization to the ER, a brief discussion addressing why the ER may be excluded from the BV trafficking network observed here would improve conceptual completeness.

Reviewer's Responses to Questions

Experimental quality

Does each figure have the proper controls?

If 'No', please indicate reasons in Comments for Author box below.

Reviewer #1:

- Yes

Reviewer #2:

- Yes
-

Were the data analyzed using appropriate statistical tests?

If 'No', please indicate reasons in Comments for Author box below.

Reviewer #1:

- Yes

Reviewer #2:

- Yes
-

Reproducibility

Were experiments performed using adequate number of biological replicates?

If 'No', please indicate reasons in Comments for Author box below.

Reviewer #1:

- Yes

Reviewer #2:

- Yes
-

Does the methods section provide sufficient detail to permit reproducibility?

If 'No', please indicate reasons in Comments for Author box below.

Reviewer #1:

- No

Reviewer #2:

- Yes
-

Completeness

Are the manuscript's conclusions supported by the data?

If 'No', please indicate reasons in Comments for Author box below.

Reviewer #1:

- Yes

Reviewer #2:

- Yes
-

Scholarship

Do the authors cite and discuss the merits of data that would argue for and against their conclusion?

If 'No', please indicate reasons in Comments for Author box below.

Reviewer #1:

- Yes

Reviewer #2:

- Yes

Does the manuscript title & abstract accurately reflect the contents of the manuscript, without hyperbole?

If 'No', please indicate reasons in Comments for Author box below.

Reviewer #1:

- Yes

Reviewer #2:

- Yes

First revision

Author response to reviewers' comments

Our response to Reviewer 1:

In this study the authors examine the intracellular distribution of biliverdin (BV) in plant cells by selectively targeting a fluorescent protein that requires BV to fluoresce to different organelles. Through this elegant technique, the authors convincingly demonstrate the presence of BV in plant cells, and the ability of BV synthesized in some organelles to be transported to others via the cytosol. The demonstration that selective targeting of enzymes that increase or decrease the levels of BV to specific organelles can influence smURFP fluorescence in other organelles is strong evidence for their model of inter-organelle BV transport. Importantly, the authors also adequately address potential complications/alternative explanations of using smURFP as a BV sensor in the discussion. I have a few suggestions that I believe would help improve the clarity and reproducibility of this manuscript.

(Q1) Most figure legends mentions a "bright field" image, but I do not see one in any of the micrographs. Including BF images would help orient readers who are unfamiliar with the system being used.

(A1) We apologize for our misdescription in figure legends. According to the comment, we added bright field images in revised figures.

Fig. 1B (an example)

Fig. 1B legend (an example)

(B, D, G) Bars, 20 μm . BF, bright field. Cell shapes are traced with dashed lines in BF images.

(Q2) Most figures include the number of cells analyzed but do not mention the number of true biological replicates. This information is necessary to understand the reproducibility of the findings.

(A2) In the revised manuscript, we added the reproducibility data in Supplementary Figs. S10 and S11. We also included the detailed information on sample size and experimental reproducibility.

Supplementary Fig. S10

Supplementary Fig. S10 legend

Supplementary Fig. S10. Reproducibility of quantification in Figs. 1-4. Reproducibility data corresponding to Figs. 1E, 1H, 2E, 2H, 3C, 3F, 4G, and 4J are shown (two independent datasets for each panel).

Supplementary Fig. S11

Supplementary Fig. S11 legend

Supplementary Fig. S11. Reproducibility of quantification in Supplementary Figs. S2, S3, and S6. Reproducibility data corresponding to Supplementary Figs. S2C, S3C, and S6C are shown (two independent datasets for each panel).

Method

Quantification of BV levels

The fluorescence intensities of smURFP and sfGFP were measured from the TIFF files using ImageJ Fiji (Schindelin *et al*, 2012). According to the bilirubin quantification method (Ishikawa *et al*, 2023), BV levels were calculated using the following formula:

$$\text{BV Level} = \frac{1}{n} \sum_{i=1}^i \left(x_i \frac{\bar{y}}{y_i} \right)$$

Where x represents the smURFP measurement value, y represents the sfGFP measurement value (Ishikawa *et al*, 2023), and \bar{y} represents the mean sfGFP intensity. Fluorescence intensities from 25 regions of five onion cells (five regions each) were measured. The experiments were repeated at least three times to confirm reproducibility (Supplementary Figs. S10 and S11). In each experiment, 1-3 onion bulbs were used.

Figure 1 legend

Fluorescence intensities from 25 plastids of five onion cells (five plastids each) were measured. Values are means \pm standard deviations ($n = 25$). Asterisks indicate significant differences (Student's *t*-test, **** $p < 0.0001$).

Figure 2 legend

Fluorescence intensities from 25 peroxisomes of five onion cells (five peroxisomes each) were measured. Values are means \pm standard deviations ($n = 25$). Asterisks indicate significant differences (Student's *t*-test, ** $p < 0.01$ [E] and **** $p < 0.0001$ [H]).

Figure 3 legend

Fluorescence intensities from 25 peroxisomes (C) or plastids (F) of five onion cells (five peroxisomes or plastids each) were measured. Values are means \pm standard deviations ($n = 25$). Asterisks indicate significant differences (Student's *t*-test, **** $p < 0.0001$ [C] and * $p < 0.05$ [F]).

Figure 4 legend

Fluorescence intensities from 25 plastids (G) or peroxisomes (J) of five onion cells (five peroxisomes or plastids each) were measured. Values are means \pm standard deviations ($n = 25$). Asterisks indicate significant differences (Student's *t*-test, **** $p < 0.0001$).

Supplementary Fig. S2 legend

Fluorescence intensities from 25 peroxisomes of five onion cells (five peroxisomes each) were measured. Values are means \pm standard deviations ($n = 25$). Asterisks indicate significant differences (Student's *t*-test, **** $p < 0.0001$).

Supplementary Fig. S3 legend

Fluorescence intensities from 25 plastids of five onion cells (five plastids each) were measured. Values are means \pm standard deviations ($n = 25$). ns, No significant difference (Student's *t*-test, $p > 0.05$).

Supplementary Fig. S5 legend

Fluorescence intensities from 50 plastids of five seedlings (10 plastids each) were measured. Values are means \pm standard deviations ($n = 50$). Asterisks indicate significant differences (Student's *t*-test, **** $p < 0.0001$).

Supplementary Fig. S6 legend

Fluorescence intensities from 25 peroxisomes of five onion cells (five peroxisomes each) were measured. Values are means \pm standard deviations ($n = 25$). ns, no significant difference (Student's *t*-test, $p > 0.05$).

(Q3) The magenta panels in most figures are difficult to see. While not necessary to change, it would be better to use grayscale images of the individual fluorescent channels and color images for the merged panels.

(A3) Thank you for the advice. If the use of grayscale images for the individual fluorescent channels, it seems to be difficult to understand the color merged panels. Based on this advice and our consideration, we added grayscale images of the individual fluorescent channels as supplementary figures (Supplementary Figs. S8 and S9) in the revised manuscript. We also added a sentence in the Method section.

Supplementary Fig. S8

Supplementary Fig. S8 legend

Supplementary Fig. S8. Grayscale images of sfGFP and smURFP in Figs. 1-4. Grayscale images corresponding to Figs. 1B, 1D, 1G, 2B, 2D, 2G, 3B, 3E, 4C, 4D, 4F, and 4I are shown.

Supplementary Fig. S9

Supplementary Fig. S9 legend

Supplementary Fig. S9. Grayscale images of sfGFP and smURFP in Supplementary Figs. S1- S6. Grayscale images corresponding to Supplementary Figs. S1, S2B, S3B, S4B, S5B, and S6B are shown.

Line 431

Images acquired by confocal laser scanning microscopy were exported as TIFF files. All fluorescent images in grayscale are included in Supplementary Figs. S8 and S9.

(Q4) The use of psSirius is mentioned in figure legends but not the results section.

(A4) In the revised manuscript, we added a relevant sentence in the Result section.

Line 110

The ectopic expression of HO1 or BVR in peroxisomes (psHO or psBVR) led to greater or diminished smURFP fluorescence intensity (i.e., BV levels), respectively (Fig. 2C-H). Peroxisome-targeted TagRFP-T (psTagRFP) and Sirius (psSirius) were used as controls for psHO and psBVR, respectively (Fig. 2D, E, G, H). These results suggest that BV accumulates not only in plastids but also in peroxisomes.

(Q5) There are a few duplicated references (Ishikawa and Kodama, Kochi et al.,) (A5) We apologize for our oversights. We carefully checked and corrected references in the revised manuscript.

(Q6) Figure S4 is only mentioned in the discussion

(A6) In the revised manuscript, we added a relevant sentence. Note that the previous Supplementary Figure S4 was changed to the Supplementary Figure S1 in the revised manuscript.

Line 104

We next asked whether BV accumulates in peroxisomes despite the absence of HO in these organelles. When a gene encoding sfGFP-smURFP fused to a peroxisome localization signal (PS^{sfGFP-smURFP}) (Fig. 2A) was expressed in onion epidermal cells, smURFP fluorescence was observed in

the peroxisomes (Fig. 2B). Note that in our preliminary experiments, we also detected smURFP fluorescence in the mitochondria, but not in the ER (Supplementary Fig. S1), suggesting that BV levels vary in different organelles. The ectopic expression of HO1 or BVR in peroxisomes (psHO or psBVR) led to greater or diminished smURFP fluorescence intensity (i.e., BV levels), respectively (Fig. 2C-H).

(Q7) In some of the figures it becomes difficult to track to which organelles the smURFP probe and the various enzymes are being targeted. It would help to add additional labels to the micrographs.

(A7) According to this comment, we added labels to micrographs.

Fig. 1D (an example)

(Q8) Using "relative BV level" as the y-axis label in several of the graphs (1E,H ; 2E,H and so on) is a little misleading. The actual measurement is the normalized smURFP fluorescence, which is used as a proxy for the levels of BV

(A8) As our measurement of BV level is not based on only smURFP fluorescence, "the normalized smURFP" seems not to be appropriate as the y-axis label. To avoid misleading in the revised manuscript, we added relevant sentences in Figure legends.

Figure 1

(E, H) BV levels were quantified using the fluorescence intensities of smURFP and sfGFP (see Methods). The mean BV level in plastids accumulating ptTagRFP was set to 1, and the relative BV level in plastids accumulating ptHO (E) or ptBVR (H) was determined.

Figure 2

(E, H) BV levels were quantified using the fluorescence intensities of smURFP and sfGFP (see Methods). The mean BV level in peroxisomes accumulating psTagRFP (E) or psSirius (H) was set to 1, and the relative BV level in peroxisomes accumulating psHO (E) or psBVR (H) was determined.

Figure 3

(C, F) BV levels were quantified using the fluorescence intensities of smURFP and sfGFP (see Methods). The mean BV level in cells expressing psTagRFP was set to 1, and the relative BV level in cells expressing ptHO (C) or psHO (F) was determined.

Figure 4

(G, J) BV levels were quantified using the fluorescence intensities of smURFP and sfGFP (see Methods). The mean BV level in cells expressing csTagRFP was set to 1, and the relative BV level in cells expressing csHO was determined.

Supplementary Fig. S2

(C) Lower relative BV levels in peroxisomes of cells expressing ptBVR. BV levels were quantified using the fluorescence intensities of smURFP and sfGFP (see Methods). The mean BV level in ptTagRFP-expressing cells was set to 1, and the relative BV level in ptBVR-expressing cells was determined.

Supplementary Fig. S3

(C) Comparable relative BV levels in plastids of cells expressing psBVR. BV levels were quantified using the fluorescence intensities of smURFP and sfGFP (see Methods). The mean BV level in psSirius-expressing cells was set to 1, and the relative BV level in psBVR-expressing cells was determined.

Supplementary Fig. S5

(C) Relative BV levels in plastids of WT and *hy2-1* cells. BV levels were quantified using the fluorescence intensities of smURFP and sfGFP (see Methods). The mean BV level in WT cells was set to 1, and the relative BV level in *hy2-1* cells was determined.

Supplementary Fig. S6

(C) Relative BV levels in peroxisomes of cells expressing ptHY2 with ptFd2. BV levels were quantified using the fluorescence intensities of smURFP and sfGFP (see Methods). The mean BV level in ptSirius-expressing cells was set to 1, and the relative BV level in ptHY2-expressing cells was determined.

(Q9) The discussion and model figure could benefit from a further examination of the transport of BV. The model could use additional explanatory text in the figure legend. Do the red arrows indicate active transport? The authors reference the active transport via ABC transporters identified by Shum *et al.*, 2021, but exogenous BV added to cultured cells has been shown to enhance smURFP fluorescence, implying that BV must be able to cross the PM of some cells via diffusion or import (Shemetov *et al.*, 2017). Could some of these findings be explained by certain organelles membranes are more/less BV-permeable due to lipid as well as protein content?

(A9) Thank you very much for the valuable suggestion. Because BV is hydrophilic, it is impermeable to biological membranes including the plasma membrane and organelle membranes. At the plasma membrane, BV is therefore considered to be transported via unknown pathways such as specific transporters or endocytosis. In the case of organelles, BV transport is likely mediated by specific transporter proteins. We revised relevant sentences in the revised manuscript. We also revised the model illustration shown in Fig. 5. In the revised model, we added transporters, which are possibly localized at the chloroplast and peroxisome, respectively.

Line 273

Because BV is hydrophilic, it is impermeable to biological membranes. Nevertheless, exogenous BV has been shown to be transported into both animal cells (HeLa cells) and plant cells (*Arabidopsis*) (Shemetov *et al.*, 2017; Parks *et al.*, 1991) via unknown pathways including specific transporters and/or endocytosis. Consistent with this, although BV moves among the plastids, peroxisomes, and cytosol (Fig. 5), its hydrophilicity prevents its passive diffusion across organelle membranes; instead, its membrane transport appears to require specific transporter proteins. In human cells, the ATP-binding cassette (ABC) transporter ABCB10 on the inner mitochondrial membrane (Shum *et al.*, 2021) exports BV from the mitochondria to the cytosol (Shum *et al.*, 2021). Given the discovery of ABCB10 in animal cells, we speculate that specific ABC transporters may also transport BV between organelles in plant cells (Fig. 5), but this possibility remains to be tested.

Fig 5

Our response to Reviewer 2:

This manuscript reports a great imaging-based strategy to visualize inter- organelle metabolite trafficking in plant cells using a smURFP (small ultra-red fluorescent protein)-based biliverdin (BV) sensor. By combining compartment- targeted smURFP with enzymatic manipulation of BV biosynthesis and degradation, the authors establish a "trans-organelle assessment" framework that enables dynamic inference of metabolite movement between plastids and peroxisomes. The work provides a technically sound proof of concept for probing organelle-level metabolite exchange and represents a meaningful methodological advance. However, some conceptual and interpretative issues require clarification to strengthen the physiological relevance and specificity of the conclusions.

Major comments

(Q1) Innovation and experimental design

The purposing of smURFP as a genetically encoded reporter for endogenous BV in plant cells is technically creative and well executed. The use of onion epidermal cells to minimize chlorophyll autofluorescence is appropriate and strengthens signal interpretation. The "trans-organelle assessment" assay wherein BV levels are artificially elevated in one organelle through heme oxygenase (HO) expression and monitored in distal compartments, which constitutes a robust experimental design that goes beyond static localization. This approach convincingly demonstrates inter-organelle metabolite mobility and could be broadly applicable to other metabolic systems.

(A1) We appreciate your deep understanding of our work.

(Q2) Bidirectional versus physiological trafficking

The data indicate that BV trafficking between plastids and peroxisomes is bidirectional under conditions of artificially elevated BV, whereas under basal conditions the dominant direction appears to be plastid-to-peroxisome. This distinction is important but currently underdeveloped. The authors propose that peroxisome-to-plastid trafficking occurs only when BV levels exceed a certain threshold. Can the authors discuss plausible physiological or stress-related scenarios (e.g., oxidative stress, heme overload, altered tetrapyrrole flux) that might generate such elevated BV concentrations in peroxisomes *in vivo*?

(A2) Thank you for the helpful suggestion. We speculate that in peroxisomes, BV is converted to bilirubin, a strong antioxidant, through a reaction with NADPH (Ishikawa et al. 2023) and/or that BV functions as an antioxidant. Elevated BV concentrations seem to be important for maintaining cellular redox homeostasis. We expect that BV trafficking from peroxisomes to plastids when BV levels increase in peroxisomes is a feedback signal to indicate a sufficient abundance of BV in peroxisomes to the BV synthetic pathway in plastids. In the revised manuscript, a new paragraph was added in Discussion section.

Line 192

Based on our data using *trans*-organelle assessment, BV trafficking from plastids to peroxisomes appears to be a constitutive process, whereas trafficking from peroxisomes to plastids is conditional, occurring upon an increase in BV levels in peroxisomes (Fig. 3D-F; Supplementary Fig. S3). As in the case of plastids (Ishikawa *et al.*, 2023), BV in peroxisomes appears to play a role in redox homeostasis through its conversion to bilirubin (a strong antioxidant) and/or its own antioxidant activity. We expect that BV trafficking from peroxisomes to plastids serves as a feedback mechanism indicating that a sufficient abundance of BV has been achieved in peroxisomes for the BV synthetic pathway in plastids.

(Q3) Specificity of smURFP for BV versus phytochromobilin

The manuscript appropriately acknowledges that smURFP may also bind phytochromobilin (PΦB), given its structural similarity to known ligands. The authors argue that PΦB is unlikely to accumulate in dark-grown onion bulb scales and further support this claim with negative results from HY2 and ferredoxin co-expression. While these controls are informative, have the authors considered validating probe specificity in a tissue or genetic background where PΦB accumulation is expected (e.g., light-grown tissues or HY2-overexpressing systems)? Such validation would more definitively establish the extent of smURFP cross-reactivity in planta.

(A3) Thank you very much valuable suggestion. Although we sought to understand whether smURFP binds to PΦB or not, it was difficult to conduct the necessary experiments. In the submitted version, we tested co-expression of HY2 and ferredoxin, but this experiment did not work in onion bulb scale cells. Because emission spectrum of smURFP is very close to that of chlorophyll fluorescence, we cannot use light-grown tissues with developed chloroplasts as a material. However, in response to the reviewer's comment, we employed dark-grown hypocotyls of *hy2* Arabidopsis mutant. Interestingly, in *hy2* mutant, smURFP fluorescence was increased in the plastid. However, further studies are needed to clarify the specificity. In the revised manuscript, the data was added as Supplementary Fig 5 and relevant sentences were also added.

Supplementary Fig 5

Supplementary Fig 5 legend

Supplementary Fig. S5. smURFP fluorescence in Arabidopsis wild-type and *hy2-1* mutants. (A) Illustration of an experiment with PT^{sfGFP}-smURFP in the plastid. Probe, sfGFP-smURFP. (B) Representative images of sfGFP and smURFP in the plastid of wild-type (WT; Col-0) and *hy2-1* mutant cells expressing PT^{sfGFP}-smURFP probe. The *hy2-1* mutant line (Kohchi *et al.*, 2001) was obtained from the Arabidopsis Biological Resource Center (CS2068). Arabidopsis seeds were surface sterilized and sown on half-strength Murashige and Skoog medium supplemented with 0.5% (w/v) gellan gum and 1% (w/v) sucrose. For germination, the seeds were incubated in the dark for 3 days

at 4°C and then exposed to continuous white light ($50 \mu\text{mol m}^{-2} \text{s}^{-1}$) for 24 h. Seedlings were then grown vertically in the dark for 3-5 days at 22°C. The resulting dark-grown seedlings were subjected to particle bombardment. Bars, 20 μm . BF, bright field. Cell shapes are traced with dashed lines in BF images. (C) Relative BV levels in plastids of WT and *hy2-1* cells. BV levels were quantified using the fluorescence intensities of smURFP and sfGFP (see Methods). The mean BV level in WT cells was set to 1, and the relative BV level in *hy2-1* cells was determined. Fluorescence intensities from 50 plastids of five seedlings (10 plastids each) were measured. Values are means \pm standard deviations ($n = 50$). Asterisks indicate significant differences (Student's *t*-test, **** $p < 0.0001$).

Line 201

smURFP is known to bind to phycocyanobilin as well as BV as a fluorophore (Rodriguez *et al*, 2016). Land plants do not produce phycocyanobilin but instead biosynthesize P Φ B, which has a similar molecular structure. Given this structural similarity, smURFP might also bind to P Φ B. Although we confirmed BV-induced smURFP fluorescence using related enzymes (HO and BVR) (e.g., Figs. 1 and 2), it is possible that the current smURFP-based imaging technique could detect P Φ B as well as BV in plant cells. However, in dark-grown *Arabidopsis hy2-1* mutants defective in P Φ B synthase (Kohchi *et al*, 2001), smURFP fluorescence was increased in the plastid, compared to wild-type cells (Supplementary Fig. S5), suggesting that smURFP does not bind to P Φ B. Alternatively, in wild-type cells, P Φ B may be rapidly exported from the plastid to the cytosol (Supplementary Fig. S5); therefore, further studies are needed to clarify the specificity. Nonetheless, we consider it unlikely that P Φ B was detected by smURFP in epidermal cells of onion bulb scales in the present study. There are...

(Q4) Speculation on BV transport mechanisms

The authors speculate that ATP-binding cassette (ABC) transporters may mediate BV translocation across organellar membranes, drawing analogy to human ABCB10. While this hypothesis is reasonable given the hydrophilic nature of BV, the discussion remains overly general. We suggest authors could strengthen this section by briefly highlighting which *Arabidopsis* ABC transporter subfamilies or members (based on predicted subcellular localization or known tetrapyrrole-associated functions) represent the most plausible candidates for future investigation.

(A4) According to this comment, we predicted subcellular localization of all ABC transporters from *Arabidopsis in silico*. We obtained amino acid sequences of *Arabidopsis* 132 proteins (Verrier *et al*. 2008 Trends Plant Sci) from UniProt database (<https://www.uniprot.org/>) and predicted their subcellular localizations and topologies using three different programs: WoLF PSORT (<https://wolfpsort.hgc.jp/>), DeepLoc (<https://services.healthtech.dtu.dk/services/DeepLoc-2.1/>), and Deep TMHMM (<https://services.healthtech.dtu.dk/services/DeepTMHMM-1.0/>). Among 132 proteins, some proteins were predicted to localize to plastids or peroxisomes. Relevant sentences were revised, and Supplementary Table 1 was added.

Line 284

Compared with the human genome (48 genes) (Dean *et al*, 2001), plant genomes contain many more ABC transporter genes: 132 in *Arabidopsis*, 261 in soybean (*Glycine max*), and 128 in rice (*Oryza sativa*) (Verrier *et al.*, 2008; Mishra *et al*, 2019; Schulz & Kolukisaoglu, 2006). When subcellular localizations and topologies of the 132 *Arabidopsis* ABC transporters were predicted *in silico*, most proteins were predicted to localize to the plasma membrane, but some proteins were predicted to localize to organelles, including plastids and peroxisomes (Supplementary Table S1). It will be necessary to develop a screening strategy for identification of potential BV transporters in various plant organelles.

Supplementary Table S1 (only part of the table is shown)